# Large-scale capsid-mediated mobilisation of bacterial genomic DNA in the gut microbiome

Tatiana Borodovich[1], Colin Buttimer [ID][1], Jason S. Wilson[2], Pavol Bardy [ID][3], Muireann Smith[1], Conor Hill [ID][1], Ekaterina V. Khokhlova[1], Matthew Harte[1], Bianca Govi[1], Paul C. M. Fogg [ID][2], Colin Hill [ID][1] & Andrey N. Shkoporov [ID][1] ✉

Transducing bacteriophage and gene transfer agents (GTAs) are constrained by the structural limits of their capsids, which determine the maximum length of host DNA they can package. Here, we utilise nanopore sequencing of intact, capsid-packaged DNA molecules to recover full-length reads, thereby enabling the precise identification of encapsidated DNA and its bacterial origin. This approach was validated using well-characterised transducing systems and subsequently applied to faecal viromes from three healthy donors. Our analysis reveals that bacterial DNA encapsidation is widespread in the gut microbiome, with up to 5.4% of capsid-packaged DNA derived from bacterial genomes. Generalised transduction and GTA activity were especially prominent in *Oscillospiraceae* and *Ruminococcaceae* (e.g. *Faecalibacterium* spp.), while lateral transduction was observed in *Bacteroides*. Additionally, we detected induction of prophages in several highly prevalent gut bacterial taxa. These findings reveal the prevalence of bacterial DNA packaging via virus or virus-like capsids in the human gut, shedding light on the diverse mechanisms that drive this process.

High taxonomic diversity[1] and functional redundancy[2] are key features of the human gut microbiome and other complex microbial communities. These attributes are thought to underpin the resilience of the microbiome to rebound to a functionally similar (if not identical) state following significant disturbances, such as abrupt dietary changes, disease-induced shifts in the gut environment, or antibiotic treatment[3,4]. Recent evolutionary models suggest that horizontal gene transfer (HGT), including phage-mediated mechanisms, may alleviate diversity constraints among competing microbial species and strains by facilitating the exchange of fitness-related genes and promoting a state of 'dynamic neutrality'[5]. Consistent with this, large-scale gut microbiome genomics and metagenomics studies reveal widespread recent horizontal gene acquisitions in bacterial genomes[6]. HGT appears to be pervasive in industrialised populations[7], with a number of these events attributed to phage transduction. However, despite extensive evidence of HGT in the gut microbiome, direct, high-resolution detection of ongoing transduction events, particularly at the level of individual particles, remains methodologically elusive.

Transduction broadly refers to any HGT mechanism in which DNA is encapsulated within a nanoparticle, such as a proteinaceous virus capsid or lipid vesicle, and transferred from a donor to a recipient cell[8]. In this context, however, we use the term transduction specifically to describe the transfer of DNA via a virus or virus-like capsid, encompassing several mechanistically distant forms. Specialised transduction (ST) occurs when a fragment of bacterial DNA adjacent to a prophage is inadvertently co-packaged with phage DNA during excision. Generalised transduction (GT) involves the packaging of random bacterial DNA fragments, often covering extensive genomic regions and, in some cases, nearly the entire genome. This process is typically initiated by phage terminase enzymes recognising pseudo-packaging

[1]APC Microbiome Ireland & School of Microbiology, University College Cork, Cork, Ireland. [2]Department of Biology, University of York, York, UK. [3]York Structural Biology Laboratory, Department of Chemistry, University of York, York, UK. ✉e-mail: andrey.shkoporov@ucc.ie

start (pseudo-*pac*) sites within the host chromosome. A recently described mechanism, lateral transduction (LT), occurs when a prophage initiates replication while it is integrated into the host chromosome, before excision. This leads to the packaging of large, contiguous regions of host DNA, initiated at one end of the integrated prophage and proceeding in fragments defined by the phage's headful packaging limit (i.e. capsid capacity)[9]. Temperate phages and phage-inducible chromosomal islands (PICI) capable of LT have been described in *Staphylococcus aureus*[9–11], *Enterococcus faecalis*[12] and *Salmonella enterica*[13,14].

Gene transfer agents (GTAs) are small phage-like particles that have independently arisen in multiple bacterial lineages through 'domestication' of ancestral phages[15,16]. Unlike true viruses, GTAs are encoded by multiple operons dispersed throughout the host bacterial genome and lack the genetic machinery for autonomous replication[17]. Their DNA packaging machinery resembles that of phage terminase complexes; however, GTA-associated terminases are typically non-specific and can package random fragments of cellular DNA[18]. To date, only a limited number of GTAs systems have been studied in detail, predominantly within *Alphaproteobacteria* such as *Rhodobacter*, *Dinoroseobacter, Bartonella* and *Caulobacter*[19].

The bioinformatic identification of novel GTA-like elements remains challenging due to their close resemblance to defective pro-phages and remnants of prophages. Unlike transducing temperate phages, characterised GTAs, such as those described in *Rhodobacterales*[20–22], *Desulfovibrio*[23], *Brachyspira*[24], *Bartonella* spp[25], *Caulobacter*[26], and *Methanococcus voltae*[27] do not preferentially package their genetic loci. Instead, they typically process and encapsulate random fragments of the host chromosome in a quasi-non-specific manner. This lack of self-DNA enrichment within GTA particles presents an additional obstacle for the reliable identification of the genes responsible for their production.

Capturing ongoing transduction events in the gut microbiome and other similarly complex microbial environments remains a significant challenge, particularly when such events occur between members of the same species. Substantial amounts of bacterial DNA have been observed in association with virus-like particles (VLPs) in gut virome studies[28]. However, these signals are often attributed to contamination[29]. When short-read shotgun sequencing is used, and no information is available regarding the size or topology of DNA molecules, it becomes difficult to distinguish genuine transducing particles from free bacterial DNA released during cell lysis. The challenge is further compounded by potential co-purification of membrane vesicles, some of which are capable of carrying DNA and mediating vesicle-based gene transfer[30–32]. One approach developed to address this issue, known as 'transductomics', involves mapping VLP-derived sequencing reads to metagenome-assembled bacterial contigs or genomes from the same sample. This can reveal coverage patterns that resemble those generated by known phage-mediated transduction mechanisms[12]. However, the resolution of this method is limited. Short DNA reads are analysed in bulk rather than at the level of individual DNA molecules, and the resulting coverage patterns can often be ambiguous, potentially matching multiple mechanisms or none of the established ones.

The length of phage-encapsidated DNA is primarily determined by the phage's DNA packaging mechanism and the physical capacity of the capsid[33,34]. Tailed phages that use sequence-specific packaging, such as those with cohesive ends (for example, *E. coli* phage λ with *cos* sites) or with terminal direct repeats (short in *E. coli* phage T7 and long in T5), produce DNA molecules of uniform and precise length. In contrast, phages that use the 'headful' packaging mechanism, which initiates at a *pac* site on concatemeric DNA (such as in *Salmonella* phage P22 and *E. coli* phage P1), generate terminally redundant, circularly permuted DNA fragments that are typically 2–10% longer than the unique genome sequence. This mechanism generally allows for a

packaging imprecision of approximately ±2%. Phage Mu and related phages combine replicative transposition with a distinctive headful packaging process, resulting in the inclusion of 500–3000 bp of host DNA along with the phage's own 37 kbp genome. Therefore, regardless of the specific mechanism of transduction, the size of packaged DNA is constrained by capsid geometry and the packaging strategy. This leads to a relatively consistent DNA length for each phage type. Transducing particles formed by tailed phages capable of generalised or LT are expected to contain bacterial DNA fragments similar in size to the phage genome, typically packaged in standard virions. In contrast, GTAs from *Alphaproteobacteria* have been shown to package DNA at ~20% reduced density into oblate $T = 3$ capsids, corresponding to a packaging capacity of about four kilobase pairs[35].

To address the current limitations in detecting active transduction events within complex microbial communities, this study aims to answer three key questions: (1) Can full-sized packaged DNA molecules, both phage genomes and transducing DNA, be reliably extracted from faecal VLPs for long-read sequencing? (2) Can this approach detect DNA fragment sizes and packaging patterns consistent with known modes of transcription, such as LT or gene transfer elements? (3) Do such events occur in the human gut microbiome, and which bacterial taxa are involved? To this end, we developed a method that preserves the integrity of encapsidated DNA and enables direct Nanopore sequencing of individual VLP-derived DNA molecules using terminally ligated sequencing adaptors. We validated this strategy across several well-characterised transduction systems and applied it to faecal samples from three healthy human donors. Our findings reveal numerous ongoing chromosomal packaging events involving key gut commensals, including *Bacteroides* and *Faecalibacterium* and provide molecular evidence supporting the activity of LT and GTA-like mechanisms in the human gut.

## Results
### Discrete peaks of packaged DNA sizes can be observed in the faecal VLP fractions
To test whether discrete peaks of packaged DNA sizes can be recovered from the faecal microbiome, we performed a pilot study on three freshly collected faecal samples from three adult donors (study protocol APC055; donor codes 925, 928, 942). Faecal VLPs were extracted by filtration, concentrated by ultracentrifugation and purified using a CsCl density step gradient. The two brightest bands visible in each gradient (further referred to as 'B' for bottom and 'T' for top) were collected. After removal of free DNA by DNAse digestion, high molecular weight (HMW) encapsidated DNA was gently extracted to prevent fragmentation. Oxford Nanopore adaptor ligation-based sequencing was used to generate sequencing reads corresponding to the full length of the extracted DNA fragments (Fig. 1A). We observed that the length distribution of Nanopore reads included discrete peaks (ranging from ~4 to ~100 kb) that are likely to correspond to distinct populations of encapsidated DNA originating from different viral agents. The sizes of the DNA fragments in these peaks are consistent with the packaging capacity of small to medium-sized tailed phage genomes and were both individual and VLP fraction B or T-specific (Fig. 1B). Sample 942T shows signs of possible DNA degradation and enrichment of short fragment sizes. Other samples were dominated by peaks of discrete read lengths with little or no visible DNA degradation.

To verify the plausibility of our interpretation of the observed pattern of read length peaks, we applied the same protocol to pure cultures of known transducing/packaging bacterial or bacteriophage systems. Sequencing of encapsidated DNA extracted from a *Bacteroides intestinalis* phage crAss001[36] lysate produced reads of ~102 kb, aligning end-to-end with the phage genome. No reads aligning to the host chromosome were obtained, suggesting a lack of transducing activity in this system (Fig. S1A). *Salmonella* phage P22, propagated on *Salmonella enterica* LT2, produced 42.6 kb reads aligning to its

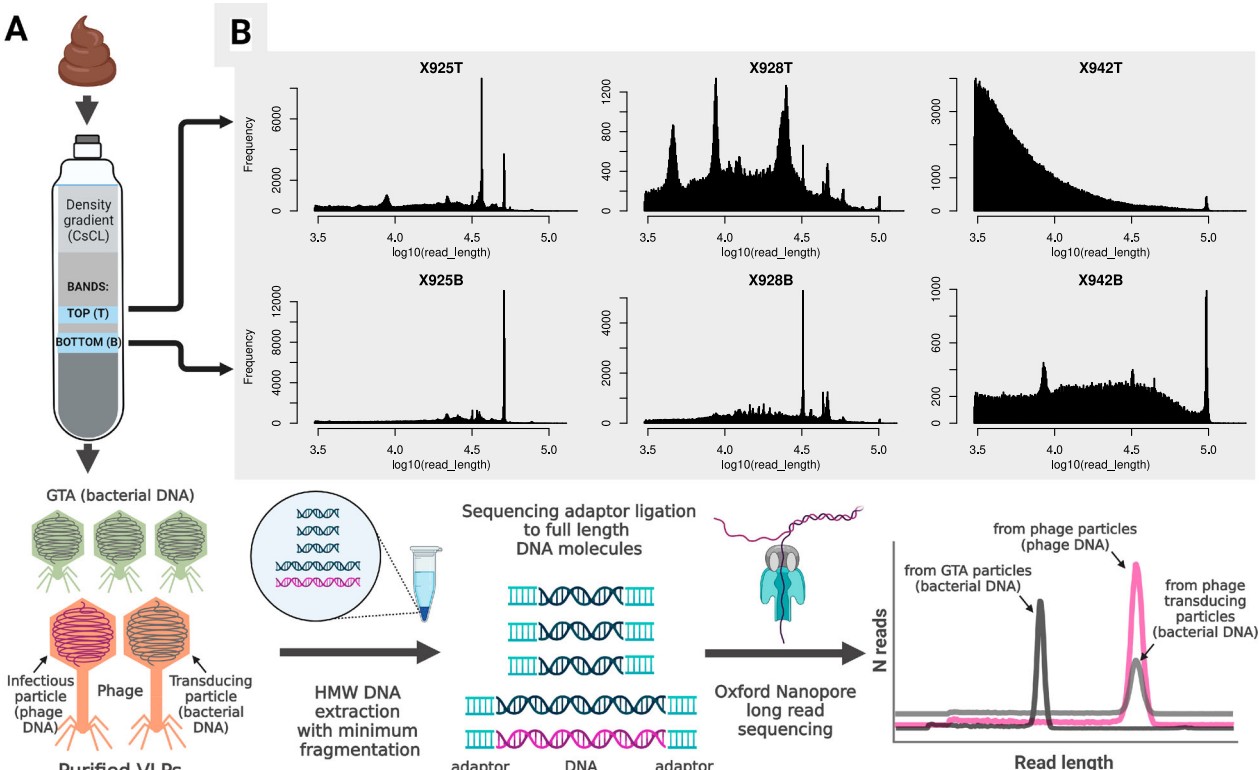

**Fig. 1 | Concept summary of the approach used in the present study. A** VLPs are purified from the human faecal samples (*n* = 3, donor codes X925, X928, X942), resulting in two VLP fractions (bands) per sample; relatively conserved DNA size packaged by each phage (both phage and bacterial DNA), and GTA leads to narrow peaks of read lengths in the Nanopore full genome-size read sequencing; image panel created in BioRender: Shkoporov, A. (2026) https://BioRender.com/28xsaxd; **B** distributions of read lengths obtained from each of the six faecal VLP fractions (-B, bottom, heavier particles; -T, top, lighter particles).

41.7 kbp genome, only slightly shorter than a commonly accepted estimate of the size of packaged DNA of 43.4 kbp[37]. Approximately 4.5% of the total number of long reads (Fig. S1B) aligned to dispersed locations in the host chromosome in a pattern consistent with a GT mechanism[12]. The *E. coli* phage P1, capable of GT, has a genome of 93.6 kbp and is expected to package 110–115 kbp of terminally redundant DNA[38]. In our hands, however, reads of 95–96 kb were obtained, with 12.8% of them aligning to various locations in the *E. coli* chromosome (Fig. S1C). A multi-lysogen strain of *Enterococcus faecalis* VE14089 was used as a reference for LT[9,12,39]. VLPs from the *E. faecalis* VE14089 culture contained one induced prophage, pp5, but also contained chromosome fragments of phage genome size adjacent to the prophage location, consistent with the LT model (Fig. S1D). Finally, we confirmed that the GTA-like element PBSX in *Bacillus subtilis* packages 13.4 kbp evenly dispersed chromosomal DNA fragments (Fig. S1E)[12,40].

These results demonstrate that the discrete DNA size peaks observed in faecal VLP fractions likely reflect packaging of diverse viral agents, including classical phages and elements with transducing potential, and highlight the utility of long-read sequencing for capturing encapsidated DNA populations in complex microbial communities.

## Taxonomic composition of packaged bacterial DNA in the faecal VLP fractions

To deeply characterise microbial and viral communities in the three donors, we reconstructed metagenomically assembled bacterial genomes (MAGs) and enabled accurate mapping of long reads by supplementing our long-read data with deep short-read Illumina sequencing. Hybrid assembly of both short Illumina reads and long Nanopore reads across all total DNA and VLP DNA samples yielded a total of 85,015, 42,751, and 45,265 non-redundant metagenomic

scaffolds (>1 kb) with a combined length of 404, 254, and 158 Mb for donors 925, 928, and 942, respectively. These numbers included 564, 423, and 201 complete and partial viral genomic scaffolds. Based on the results of metagenomic binning, assigned taxonomy (Genome Taxonomy Database, GTDB), and inducible prophage content (prophages detectable in the VLP fraction assemblies), all scaffolds were assigned to the following categories: (1) purely viral (mainly bacteriophage) scaffolds (*n* = 1137 across the three subjects); (2) un-binned bacterial scaffolds (*n* = 145,909); (3) bacterial scaffolds with inducible prophages (*n* = 135); (4) bacterial scaffolds binned into 116 MAGs (*n* = 22,731); (5) MAGs scaffolds with inducible prophages (*n* = 32); (6) viral scaffolds binned with bacterial MAGs (*n* = 51). In addition to that, 850 archaeal scaffolds were assembled, with some of them binning into two archaeal MAGs, and none containing identifiable inducible proviruses.

As expected, the sequence space (% of Illumina reads aligned) in the total community DNA was entirely dominated by bacterial MAGs and solitary scaffolds, with ≤2.7% of reads aligning to purely viral scaffolds (Fig. 2A). At the same time, VLP DNA fractions B and T mainly contained sequences mappable to either purely viral scaffolds, or bacterial scaffolds and MAGs containing prophage sequences. Nevertheless, 0.8–4.6% of reads in the B fraction and 3.6–5.4% of Illumina reads in the T fraction aligned to purely bacterial scaffolds (in or outside of MAGs), indicating either contamination with total community DNA or bacterial DNA packaged into VLPs. Notably, the family-level composition of this subset of VLP reads appeared visually distinct from that of the total community DNA (Fig. 2B). In the total community DNA, the families *Bacteroidaceae*, *Rikenellaceae*, and *Lachnospiraceae* were the most abundant. In contrast, the VLP fractions (especially fraction T and donor 942) were dominated by families *Ruminococcaceae* and UBA660 (according to the GTDB taxonomy). However, none

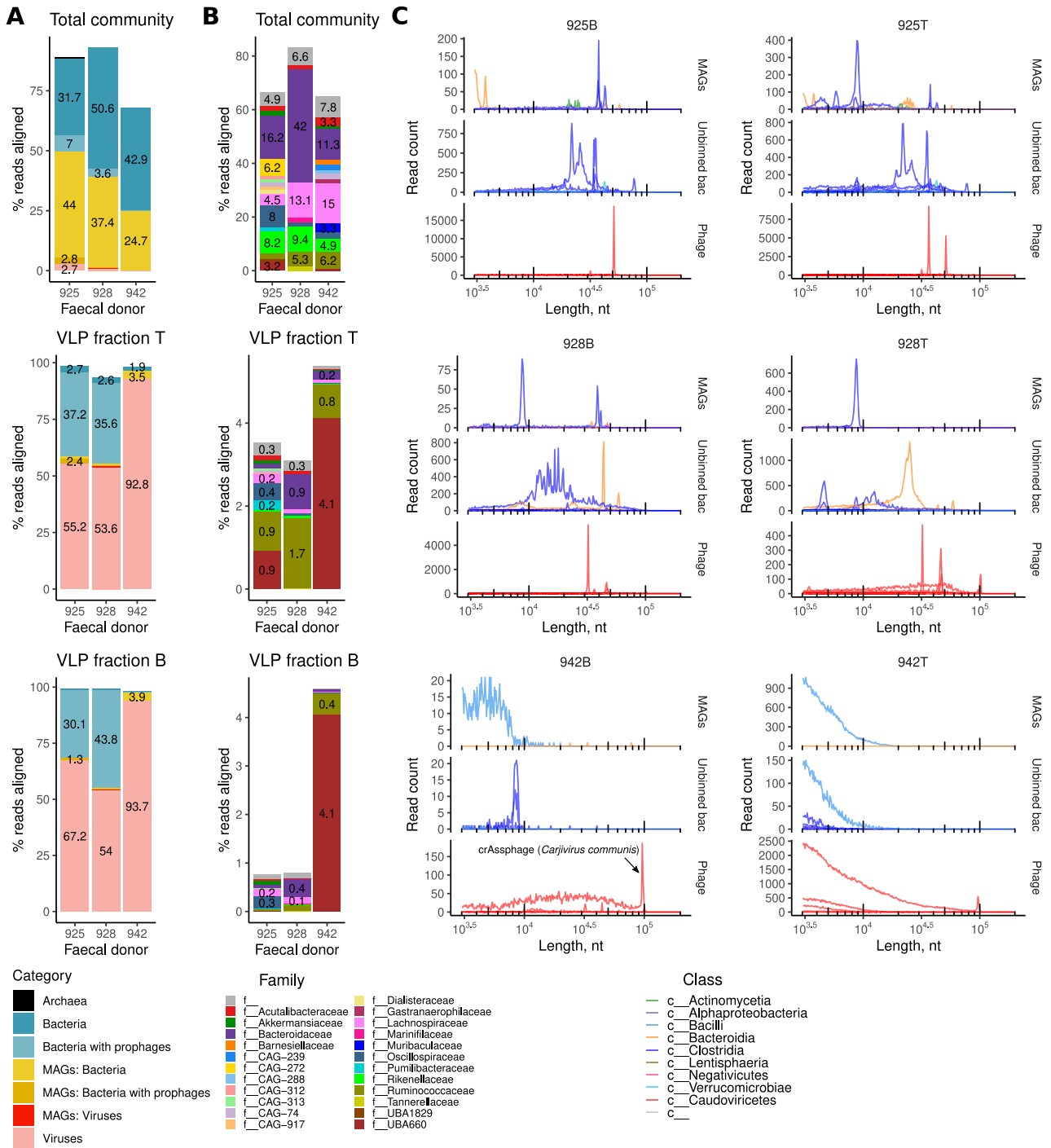

**Fig. 2 | Taxonomic composition of DNA in the faecal VLP fractions and complete genome Nanopore read sizes associated with different taxonomic groups. A** Composition of metagenomic short reads in the total faecal community DNA, as well as two VLP fractions after mapping to metagenomic scaffolds; **B** Family-level composition of bacterial DNA in the same samples; **C** Dissection of Nanopore complete genome-size read length distributions by mapping them to various metagenomic entities; each curve is a separate MAG/bacterial species/virus genome; curves are coloured according to taxonomic class (both bacteria and phages)—see text and Figs. S5–S11 for more details.

of these differences reached statistical significance ($\alpha = 0.05$) when assessed across all three donors in either the DESeq2 and ALDEx2 frameworks, likely due to small sample size and high inter-individual microbiota variation. Nonetheless, DESeq2 detected several minor taxa that were enriched in VLP fractions, such as *Propionibacteriaceae*, SC72, and UBA1402 (Fig. S2).

To further investigate the origins of this enrichment, while accounting for the high individuality of the microbiota in each of the

three donors, we examined the intra-individual ratios of the relative abundance of each genomic scaffold in the VLP fractions versus the total community DNA. The distribution of these ratios deviated from the normal distribution in the upper quantiles ($p = 1.3e-36$, Kolmogorov–Smirnov test), indicating disproportionate enrichment of some scaffolds (viral and bacterial) in the VLP fractions (Fig. S3A). We arbitrarily set a cut-off of +2 SD to select for enriched scaffolds (Fig. S3B) and extracted bacterial MAGs that contained such VLP-

enriched scaffolds. As shown in Fig. S3C, VLP fraction B primarily contained DNA mapping to individual overrepresented MAG-associated scaffolds that contained prophages (genera *Bifidobacterium*, *Dysosmobacter*, and *Ruminococcus*). At the same time, VLP fraction T provides examples of enrichment of nearly the entire bacterial MAGs belonging to genera *Gemmiger*, CAG-115 (family *Ruminococcaceae*), HGM11616 (family *Christensenellaceae*), and CAG-628 (candidate family UBA660), in line with the composition of dominant bacterial families seen in Fig. 2B.

These findings suggest that while the denser VLP fraction (B) is enriched for encapsidated viral genomes, the lighter VLP fraction (T) harbours bacterial DNA, some of which may originate from whole genome packaging events, reflecting both active prophage induction and potential GT. The taxonomic skew towards specific bacterial families distinct from the total community indicates selective DNA packaging processes rather than random contamination.

## Long-read sequencing of VLP-associated bacterial DNA reveals different transduction modes

We then proceeded to characterise the sequence space of long Nanopore reads obtained from HMW VLP DNA, considering read length distributions. We obtained 0.77 million reads greater than 3 kb in length, of which 0.7 million were mapped successfully to genomic scaffolds and assigned to either a bacterial MAG or, in the case of solitary bacterial and viral scaffolds, to the lowest taxonomic rank possible. Reads were then classified into three broad categories: bacterial MAGs (including prophage regions), unbinned bacterial scaffolds (including prophage regions), and viruses. After removing reads aligning to rare entities, 0.68 million long reads were retained, bearing 146 unique taxonomic labels across the three human donors. The overall taxonomic composition of filtered long reads from the two VLP fractions was roughly similar to that of the short read data (Fig. 2B), while showing an even higher percentage of non-phage bacterial DNA (up to 18% in the top fraction, and up to 13% in the bottom, Fig. S4A, B). As shown in Fig. 2C, reads associated with specific taxa demonstrated very specific read length distributions, often concentrating into narrow peaks (from 4.6 to 100 kb), with one or more specific narrow peaks per taxon or MAG. Interestingly, peaks representing some of the longest reads obtained in this study (~100 kb) were mapped to a nearly complete genome of crAssphage (*Carjivirus communis*, 93.1 kb), the most abundant bacterial virus in the human gut[41].

Reads produced from the two VLP fractions differed in their length, with the bottom fraction (denser particles) being associated with longer reads ($p < 2.2e{-}16$ in the Wilcoxon test). Using the edgeR statistical framework with natural spline modelling (df = 3), we explored how the abundance of microbial families changes across sequencing read lengths to detect gradual, non-linear trends. We detected several family-level taxa exhibiting significant non-linear abundance trends along the read length gradient (FDR $< 10^{-5}$). Taxa such as *Oscillospiraceae* and CAG-508 increased in representation with longer reads (~20–40 kb), while others, such as *Ruminococcaceae*, were more abundant at shorter length bins (~8 kb, Fig. S4D).

Manual examination of individual read-length distributions and peaks per each MAG and taxa, as well as examination of read mapping patterns of both the short and long reads to all scaffolds constituting a given MAG or a taxonomic unit in a given sample (Fig. S5), allowed us to identify 63 discernible packaging events involving bacterial genomic DNA (including prophages, Supplementary Data File 1, Figs. S6–S11), across all three donors. Based on read length profiles and mapping to the reference scaffolds, many packaging events associated with the VLP B fraction could be assigned to the induction of prophages (20/31). In contrast, the remaining cases were consistent with GT, LT and GTA events[12]. The majority (22/32) of events observed in the VLP T fraction were consistent with GTA-like packaging. Some examples of these events are shown in Figs. 3 and 4.

The first row of panels in Fig. 3A, B shows an example of the packaging of genomic DNA by a likely GT from an uncultured species F23-B02 sp900545805 (family *Oscillospiraceae*) in the VLP samples from donor 925. Twelve un-binned genomic scaffolds of this species (of which only one is represented in Fig. 3) with a combined length of ~2.4 Mb (Fig. S12) show very uniform coverage by both the Illumina and Nanopore reads, with the latter ones forming a narrow peak of lengths at 34.9 kb, consistent with packaging by phages previously described in this bacterial family, with typical genome sizes of 35–37 kb[42]. The scaffold shown in Fig. 3B contains a prophage of an estimated size of ~41 kb; however, no evidence of its induction or preferential packaging can be observed, suggesting that a different phage is responsible for the observed packaging. As shown in Supplementary Data File 1, additional sets of genomic scaffolds assembled from the same faecal sample were taxonomically annotated as family *Oscillospiraceae* and class *Clostridia*, exhibiting similar patterns of VLP read lengths to F23-B02 sp900545805 and therefore might originate from the same species. Some of these scaffolds contain complete phage/prophage genomes with genome sizes of 33.3–34.5 kb, producing packaged DNA of similar size (34–35 kb; Supplementary Dataset at https://doi.org/10.6084/m9.figshare.26310658). These scaffolds can be speculatively implicated in the GT event observed in the species F23-B02 sp900545805.

An example of an LT mechanism is shown in Fig. 3C–F. One of the genomic scaffolds belonging to the genus *Bacteroides sp.* in subject 928 contained a highly induced 56.9 kb prophage, producing full-length Nanopore reads (58.4 kb) with random terminal redundancies, as well as shortened reads of ~23.5 kb, mainly originating from the VLP fraction T ("top" band—less dense phage capsids; Fig. 3C, D). As expected, the majority of reads within these two length peaks align to the circularised/concatemerized phage genome. However, a significant fraction of reads within these peaks includes "chimeric" reads, starting from the hypothetical *pac* site located next to the terminase gene and extending into the bacterial chromosome outside the prophage region, as well as reads with alignment start sites (multiples of 58.4 kb) located at regular intervals from the prophage *pac* site. For instance, among reads longer than 40 kb, 1,731 aligned within the prophage, 500 aligned outside of the prophage, while 13 reads were chimeric, starting at the putative *pac* site and extending beyond the prophage boundary (Fig. 3D). This is consistent with the model of in situ prophage over-replication and unidirectional chromosomal DNA packaging that defines the LT mechanism. The shorter reads align at regular intervals in such a manner that the alignment start site coincides with the start site of the full-length reads. Still, the end site is located at a random length from the start (~23.5 kb on average), suggesting that they could have resulted from partial capsid DNA ejection and degradation after packaging, during storage, or the VLP purification steps. This agrees well with the fact that these types of reads are associated with the VLP fraction T (less dense particles). In the VLP fraction T, the shortening of packaged DNA fragments creates a periodic wave of coverage by Illumina short reads extending from the prophage site. A number of *Bacteroides* genomic scaffolds ($n = 10$, total length 2.7 Mb), the longest of them being 503 and 669 kb, were recovered from the total community assembly in subject 928 with the same pattern of coverage by Nanopore and Illumina reads (Fig-S. 3F and S1), suggesting very efficient LT-type packaging of a substantial portion of the genome, if not the entire genome, in this strain of *Bacteroides sp.* At least one additional inducible 44.2 kb prophage (packaged DNA size 43.4 kb) is likely to be part of the same genome, but it is incapable of LT (Fig. 3E).

Another example of possible LT, albeit with a lower read coverage, is presented in Fig. 3G, H. One out of the three genomic scaffolds from subject 925, belonging to a bacterium in the order *Christensenellales*, contains a 36.8 kb prophage element, with 37.4 kb Nanopore reads aligning to its circular or concatemerized isoform. A few read

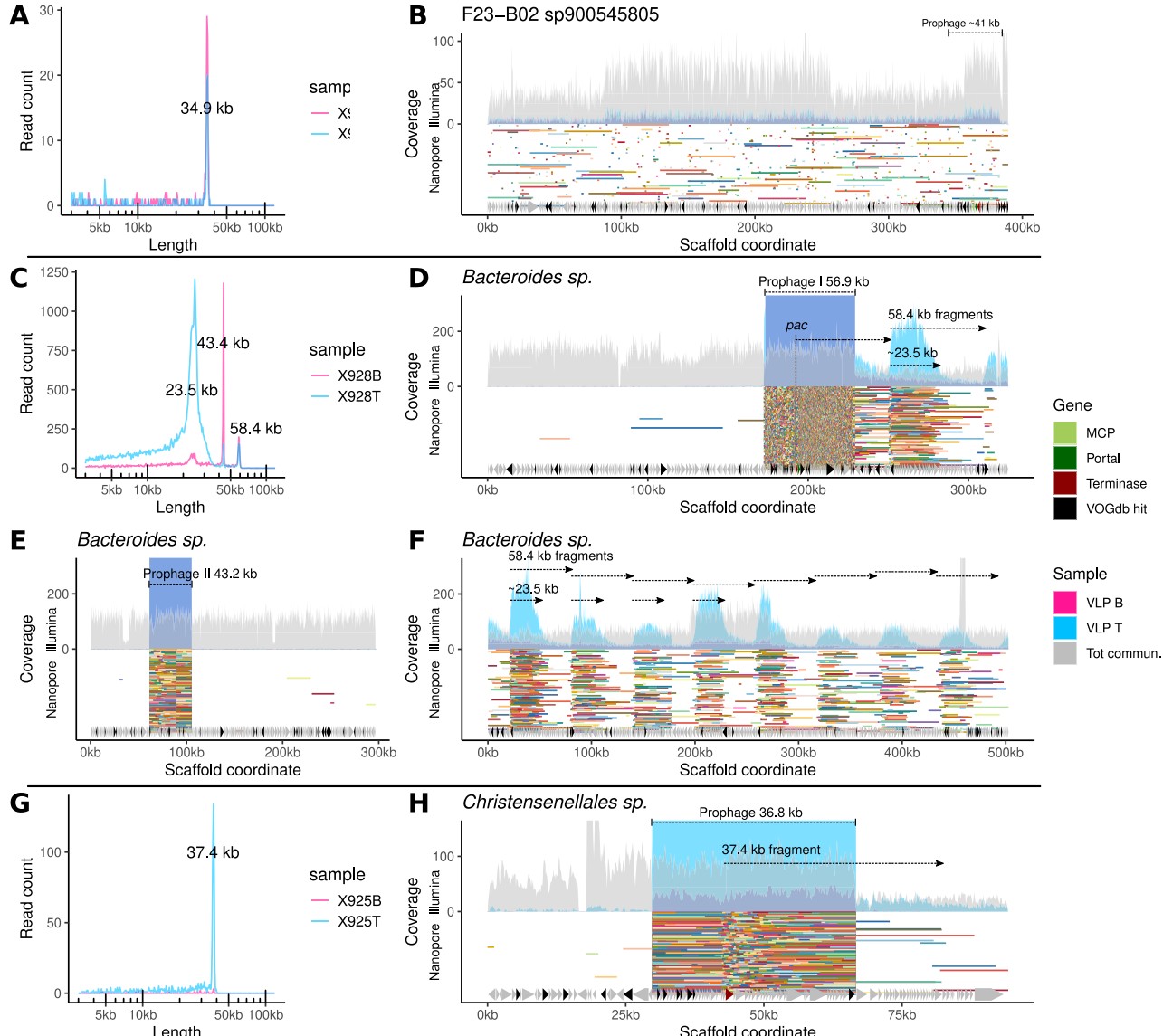

**Fig. 3 | Examples of read sizes and mapping patterns of full-sized Nanopore reads consistent with GT and LT in different gut bacteria. A, B** Distribution of full-sized Nanopore VLP DNA read lengths and their mapping to an uncultured bacterial species' F23-B02 sp900545805 389 kb genomic scaffold in donor 925 – possible GT; area plots in **B** represent coverage by short Illumina reads, whereas segments of random colours are long Nanopore reads (more scaffold are shown in Fig. S12); **C** Three peaks of VLP DNA read lengths associated with a *Bacteroides sp.* bacterium in donor 928; 58.4 kb is a full-sized read length corresponding to a 56.9 kb prophage I and transducing particle DNA (LT); 23.5 kb ones represent partial phage and bacterial DNA fragments (LT) linked with the same prophage I; 43.4 kb reads align to the second predicted 43.2 kb prophage II; **D–F** Mapping of reads from C to three metagenomic scaffolds of *Bacteroides sp.* (see text for additional interpretation and Fig. S1 for more metagenomic scaffolds from *Bacteroides sp.* in donor 928); **G, H** distribution of read sizes and mapping pattern of reads aligning to a 36.8 kb prophage and adjacent chromosomal regions in a *Christensenellales sp.* bacterium in donor 925 (possible LT).

alignments, however, begin in the vicinity of the terminase gene (a possible *pac* site) and extend into the chromosomal DNA downstream of the prophage integration site, while maintaining the same characteristic length of 37.4 kb.

Therefore, long-read sequencing of VLPs from the human faecal samples reveals that bacterial DNA is frequently packaged into phage particles through diverse transduction mechanisms, including GT, LT and GTAs. These mechanisms are taxon and sample-specific, with distinct read length signatures and mapping patterns. The presence of full-length and chimeric reads, along with differential coverage profiles, provides strong evidence of widespread and varied DNA packaging strategies in the human gut microbiome, highlighting a significant and previously underappreciated route of HGT.

## GTA-like packaging is prevalent in the gut and is primarily associated with the families *Ruminococcaceae* and *Oscillospiraceae*

The most common mechanism of VLP packaging of bacterial genomic DNA observed in this study was encapsidation of short seemingly random fragments of chromosomal DNA, with relatively even coverage of large spans of genomic scaffolds. This is consistent with the presence of active GTA elements. Examples of this type of DNA packaging were observed in the families of gram-positive anaerobic bacteria, including *Ruminococcaceae* (*Faecalibacterium* spp., *Gemmiger* spp., CAG-115, *Ruminococcus callidus*, 17/25 GTA events, Fig. 4), *Oscillospiraceae* (*Onthomonas* sp., 3/25 events), and *Christensenellaceae* (HMG11616, QANA01, 3/25 events). Possible GTA-like packaging in *Faecalibacterium* spp. was associated with several MAGs, as well as sets

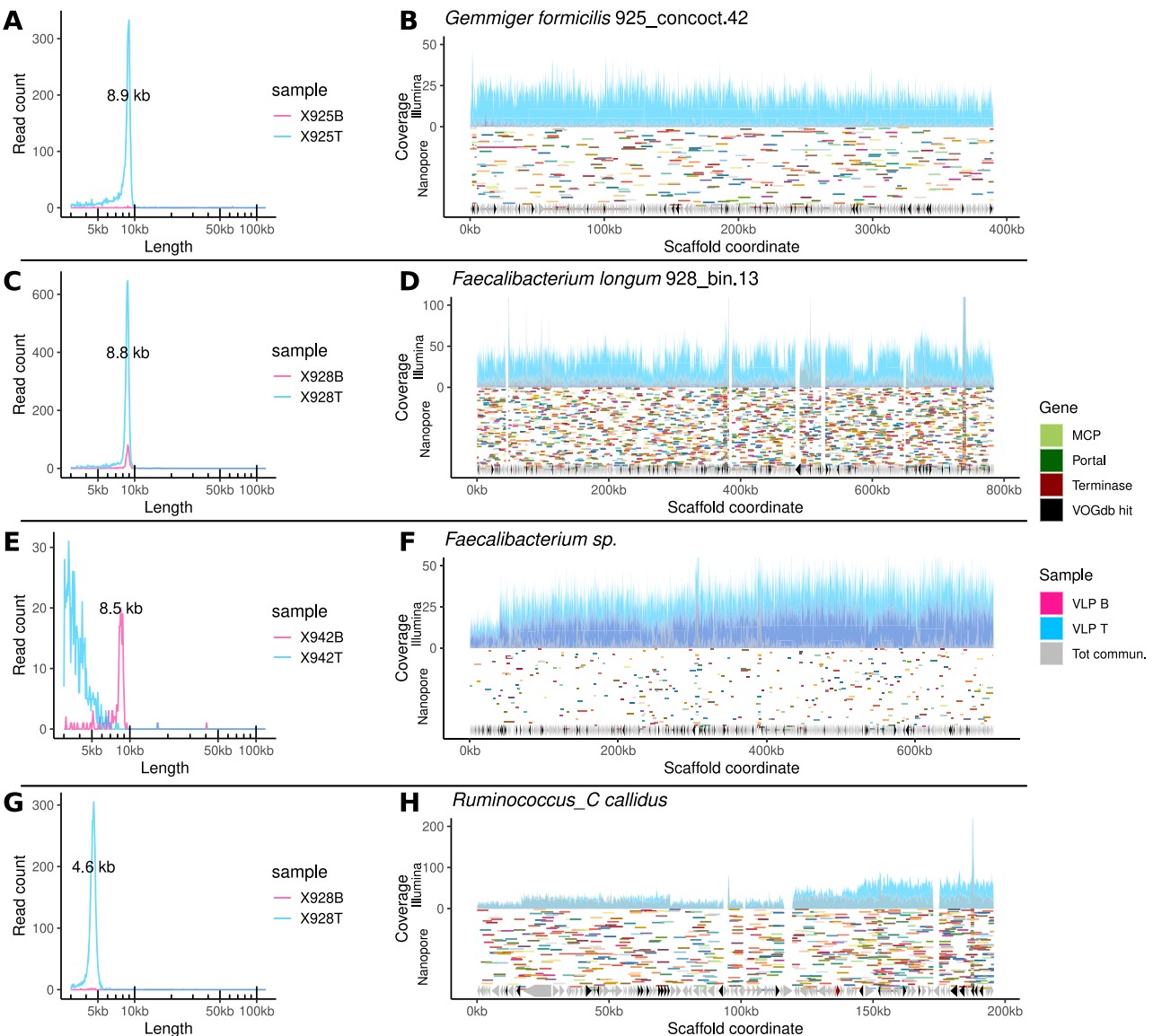

**Fig. 4 | Examples of read sizes and mapping patterns of full-sized Nanopore reads consistent with GTA-like packaging in families *Ruminococcaceae* and *Oscillospiraceae* of gut bacteria. A**, **B** Distribution of full-sized Nanopore VLP DNA read lengths and their mapping to one of the genomic scaffolds included in MAG 925_concoct.42 (*Gemmiger formicilis*) in donor 925–possible GTA; area plots in **B** represent coverage by short Illumina reads, segments of random colours are long Nanopore reads; **C**, **D** Same for one of the scaffolds in MAG 928_bin.13 (*Faecalibacterium longum*) from donor 928; **E**, **F** Same for a solitary scaffold from *Faecalibacterium* sp. in donor 942; **G**, **H** Same for a solitary scaffold of *Ruminococcus callidus* in donor 928.

of unbinned genomic scaffolds, across all three donors, consistent with the high relative abundance and prevalence of this microbial genus in the human gut microbiome. The lengths of fragments packaged in *Faecalibacterium* spp., *Gemmiger* spp. varies over a very narrow range: 8.4–8.9 kb, whereas *R. callidus* provides an example of much shorter fragment packaging of just 4.6 kb.

A search was conducted for GTA-like elements in several MAGs showing patterns of VLP long-read coverage indicative of GTA production (925_bin.12 – *Onthomonas* sp900545815; 925_concoct.42 – *Gemmiger formicilis*; 925_bin.142 – *F. prausnitzii*; 928_bin.13 – *F. prausnitzii*). One of these MAGs, 925_bin.142 contained a strong candidate for a GTA operon. The 10.5 kb operon (FpGTA I) consists of ten open reading frames, whose predicted products include a possible small terminase subunit (TerS) lacking any obvious DNA binding domain[18], as well as homologues of a large terminase (TerL), portal protein, head scaffolding and major capsid proteins, as well as several conserved phage-related hypothetical proteins. We conducted a search for homologous regions across all metagenomics scaffolds in this study, as well as in reference genomes of *Faecalibacterium*, *Gemmiger* and *Subdoligranulum* (*n* = 16). We performed clustering of all predicted protein products using MMseqs2 based on amino acid sequence similarity. Functional annotations of FpGTA genes from 925_bin.142 were then extrapolated to all members of the corresponding protein clusters (PCs). Regions with concentrations of several putative FpGTA-related PCs are shown in Fig. 5A. Putative FpGTA operons homologous to the one in 925_bin.142 are present in all twelve *Faecalibacterium* genomes examined (*F. prausnitzii* APC918/95b, APC924/119, ATCC 27768, ATCC 27766; *F. intestinale* APC923/51-1; *F. longum* APC942/30-2, APC942/18-1, APC942/32-1; *F. hattorii* APC922/41-1; *F. duncaniae* A2-165 and *Faecalibacterium* sp. APC942/8-14-2 and APC923/61-1), but not in *Gemmiger* and *Subdoligranulum*.

Anaerobic broth cultures of some of these strains were prone to lysis in the late stationary phase; for example, liquid cultures of *F. prausnitzii* ATCC 27766 could be observed to lyse after 48–72 h post-

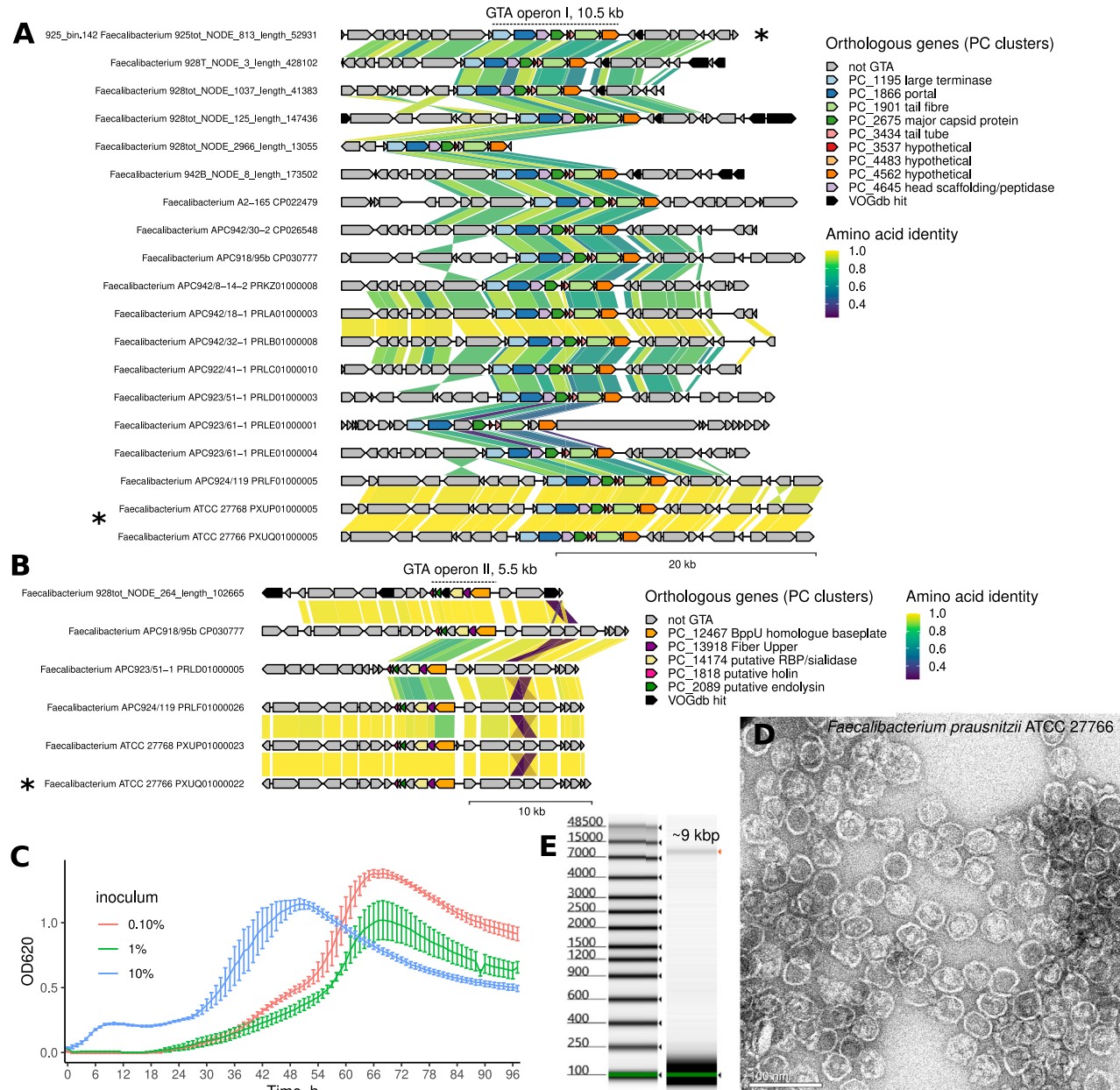

**Fig. 5 | GTA-like elements encoded in MAGs and metagenomic scaffolds from *Faecalibacterium spp*. assembled in this study, as well as in genomes from *Faecalibacterium spp*. isolates. A** A 10.5 kb candidate GTA operon I was identified of a 52.9 kb metagenomic scaffold included in *Faecalibacterium prausnitzii* MAG 925_bin.142 from donor 925; homologues of this operon are widely distributed in other genomic scaffold from *Faecalibacterium spp*., as well as in eleven genomes of *F. praunsitzii* isolates (including ATCC 27766), an *F. duncaniae* genome (A2-165) and an *F. hattorii* genome (APC922/41-1); annotated GTA-like structural protein genes are highlighted with colours; other genes with hits to Virus Orthologous Groups Database (VOGdb) are highlighted in black; link colours are based on protein-protein amino acid identity values calculated using MMseqs2; **B** A separate 5.5 kb operon encoding additional structural proteins and lysis genes identified in a *Faecalibacterium sp*. genomic scaffold, assembled in this study, as well as in several *F. prausnitzii* isolates, including ATCC 27766; **C** Growth and spontaneous lysis curves of *F. prausnitzii* ATCC 27766 in liquid media with different inoculum volumes (experiment repeated twice with similar results, three biological replicates from one of the attempts are shown, error bars are mean ± SE); **D** Transmission electron microscopy of capsid-like structures (~40 nm in diameter) spontaneously produced in broth culture of *F. prausnitzii* ATCC 27766 (performed twice with similar results); **E** Electrophoresis of DNA extracted from capsid-like structures produced by *F. prausnitzii* ATCC 27766 (performed three times with similar results).

inoculation (Fig. 5C). A culture supernatant collected from the late stationary phase of *F. prausnitzii* ATCC 27766 was concentrated and separated using CsCl step gradient ultracentrifugation, resulting in the detection of a single band (approximately corresponding to the top band of faecal VLPs, density of ~1.3 g/mL). Transmission electron microscopy (TEM) of this material revealed capsid-like particles with an approximate diameter of 40 nm (Fig. 5D), which is consistent with a potential $T = 3$ capsid architecture and packaging capacity of 8.5 kb

DNA if the DNA is packaged at a density equivalent to RcGTA and CcGTA[26,35]. Indeed, DNA extracted from the particles formed a sharp band of fragment sizes at ~9 kbp, consistent with fragment sizes of 8.4–8.9 kb observed in nanopore metagenomic sequencing of packaged *Faecalibacterium* DNA (Fig. 5E). The capsid particles have no identifiable tails. However, such structures may have been lost during purification or may be very short. Indeed, the FpGTA I cluster encodes proteins homologous to those used by short-tailed phages. LC-MS

proteomics analysis of these particles demonstrated that structural proteins encoded by the FpGTA I operon (e.g. major capsid protein RCH50849.1, and portal protein RCH50851.1) constitute the majority of all ATCC 27766 proteins present in the sample (Supplementary Data File 2). Moreover, several phage-like structural proteins encoded in the ATCC 27766 genome, outside of the FpGTA I operon, were detected. When mapped to the genome of ATCC 27766, these formed a second operon (FpGTA II, 5.5 kb), comprising structural protein genes, including those for the baseplate, tail fibre, a putative receptor-binding protein (RBP), as well as the lysis proteins—holin and endolysin. No other putative prophage proteins were detected that could account for the encapsidated DNA. Taken together, these two operons probably constitute most, if not all, of the functions needed for FpGTA particle morphogenesis[17] (most likely podovirus-like), DNA packaging and cell lysis in *F. prausnitzii* ATCC 27766. GTAs often require auxiliary structural or maturation genes, and we cannot exclude the possibility that more will be identified here. The two FpGTA loci are also present in strains APC924/119, APC923/51-1, APC918/95b, ATCC 27768, but the FpGTA locus II was not detected in the remaining seven strains. It is likely that the strains that only carry the FpGTA operon I have either lost the FpGTA operon II or possess a more divergent set of genes that were overlooked by our analysis. In the strain APC918/95b, for which a complete circular genome sequence is available (CP030777), the two operons (FpGTA I: 2536737-2546346 and FpGTA II: 2672270-2677176) are separated by -125 kbp.

These results demonstrate that GTA-like DNA packaging is widespread in the human gut microbiome, particularly among Gram-positive anaerobes, such as *Faecalibacterium* and *Ruminococcus* species from the *Ruminococcaceae* and *Oscillospiraceae* families. These taxa package short, consistent DNA fragments (4.6–8.9 kbp) into VLPs, a hallmark of GTA activity. In *F. prausnitzii*, two distinct FpGTA operons were identified, and the production of phage-like particles was experimentally validated through microscopy, proteomics, and VLP purification, supporting their function in FpGTA particle production and HGT. Thus, suggesting that GTAs are a prevalent and potentially important mechanism of gene flow in the gut ecosystem.

## Discussion
We employed a long-read single-molecule nanopore sequencing-based approach to identify capsid-mediated packaging and mobilisation of bacterial DNA in the human gut microbiome. The size of the packaged single DNA molecule adds an essential extra layer of information, allowing us to distinguish between encapsidated DNA (producing narrow peaks of read lengths, indicative of the size and type of capsids) and contaminating and degraded bacterial DNA. This approach also allows individual capsid-packaged DNA molecules to be traced, thereby enabling us to determine their origin, chromosomal location, and the structure of their ends. It also facilitates the identification of circularly permutated (headful packaging) and chimeric reads (ST and LT), and in some cases, links transducing particles with the likely origin phages. This approach is also uniquely suitable for identifying instances of host DNA encapsidation involving GTAs and GTA-like elements. GTAs can be challenging to study because the GTA-encoding genes are not selectively packaged, but rather the entire host genome is packaged relatively uniformly[12].

Our results from three human faecal samples show extensive packaging of host DNA, representing up to 5.4% of the total VLP DNA content. This was broadly representative of all main bacterial phyla present in the human gut, and included examples of possible GT, ST, LT, and GTA. Ongoing packaging events involving phyla *Bacillota* (GT, GTA) and *Bacteroidota* (LT) were especially prominent. This agrees with numerous reports providing evidence of a high rate of HGT, including phage-mediated HGT in the human gut[6,7], and its central role in the evolution of complex microbial communities on ecological time scales[43,44]. Interestingly, a recent study examining the rates of inter-

species HGT in the human gut demonstrated that class *Clostridia* (phylum *Bacillota*) and order *Bacteroidales* (phylum *Bacteroidota*) show the highest rates of HGT[45]. An earlier study demonstrated extensive HGT between *Bacteroidales* species in the human gut involving integrative and conjugative elements (ICE)[46].

Many bacterial packaging events in our dataset were prophage induction events. It has been previously shown that lysogens are prevalent in the gut bacteriome[47,48] and that temperate phages dominate in the gut virome[49,50]. Furthermore, the gut environment is conducive to prophage induction[51,52] through signals produced by the host organism, resident bacteria, and external sources (antimicrobial peptides, quorum-sensing autoinducers, immune and hormonal signalling, nutrients, bile salts, etc.)[53], which subsequently affects transduction by temperate phages and, likely, GTAs that are also inducible through the SOS response mechanism[17].

One of the interesting findings in our study was a clear signature of prophage-driven LT in *Bacteroides*, the most abundant genus of bacteria in the human gut microbiome[54]. LT was initially discovered in temperate phages of *S. aureus*[9] and subsequently in *S. aureus* PICI[11]. It was later observed in *E. faecalis*[12] and *Salmonella*[13,14], demonstrating that LT is neither unique to *Staphylococcus* nor to the phylum *Bacillota*. In this study, we expand the taxonomic range of LT-capable phages even further, with examples of LT observed in *Bacteroidales* (phylum *Bacteroidota*) and *Christensenellales* (phylum *Bacillota*).

While it is evident that phage involvement in genetic exchange can provide a benefit to the bacterial population[44,55,56], it appears to be costly to the phages, as it decreases the number of infectious particles. This raises the question of why phages often engage in host DNA mobilisation, seemingly to the detriment of their own spread in the population. One study suggests that GT confers fitness advantage to temperate phages in rapidly changing environments and can be seen as an adaptation inherent to their life cycle[57]. Transduction by temperate phages can promote adaptations and increase the survival of the lysogenized subpopulation while eliminating non-lysogenized competition, thus ensuring vertical transfer of the prophage[58]. A recent study shows that *Salmonella pac*-type phages have evolved distinct GT strategies[14], indicating that host DNA mobilisation is a consistently conserved adaptation rather than a persistent mistake.

The most striking observation was evidence of GTA operating at high efficiency in members of the families *Ruminococcaceae* and *Oscillospiraceae*, including the genus *Faecalibacterium*—another highly abundant microbial genus in the human gut. Very few GTA elements have been described so far[17,35,59–61]. In host-associated bacteria, GTA production has been observed in the human pathogen *Bartonella bacilliformis*[62], and GTA-like gene clusters were found in *Brucella abortus* and *Brucella suis* genomes[60], but not in human gut commensals. The *Faecalibacterium* GTA-like element FpGTA characterised in this study is the first such element discovered in the human gut microbiome, as well as the first GTA-like element with podovirus-like morphology, and one of the very few known GTA-like elements outside of *Alphaproteobacteria*.

Our understanding of GTA operation, prevalence and phylogeny remains extremely limited, with formal recognition and taxonomic classification of GTAs only recently accepted, and encompassing primarily GTAs in *Alphaproteobacteria* and *Spirochaetia* (*Brachyspira*)[15,16]. In other phyla of bacteria and archaea, there are examples of DNA packaging consistent with the GTA model (phage-like particles packaging random fragments of host DNA with no preferential packaging of phage-like genes)[23,24,63], but the associated GTA gene clusters are yet to be identified. Importantly, FpGTA is the second known example of a GTA/GTA-like system in Gram-positive bacteria (the first being PBSX in *Bacillus subtilis*).

FpGTA production could potentially play a role in genetic exchange in *Faecalibacterium*, where FpGTA operons appear to be near-ubiquitous. Our results show that at least some of these operons

are active and produce GTA particles, but further research is needed to determine whether the particles are capable of successfully delivering their transducing DNA, or if they instead function similarly to the GTA-like element PBSX in *B. subtilis*[40,64], which does not inject packaged DNA but acts as a tailocin[65] (antibacterial protein complexes similar in morphology to phage tails that kill closely related strains via cell wall and membrane damage). If *Faecalibacterium* FpGTAs are indeed a functional transduction system, they could confer a fitness advantage beyond adaptive trait acquisition. It has been demonstrated that in *Caulobacter crescentus*, gene transfer by GTAs enhances cell survival during the stationary phase and following DNA damage, facilitating DNA repair through homologous recombination[26]. A previous comparative analysis of *Faecalibacterium* genomes demonstrated strikingly low synteny between available genomes, indicating high genome plasticity, as well as high prevalence of ICE[66]. Whether or not the frequent genetic exchange between individual strains via FpGTA contributes to high genome plasticity and low genomic synteny in the genus *Faecalibacterium*, with or without the involvement of ICE, remains to be established. However, it seems logical to propose that in a complex and dense microbial community, such as the human gut microbiome, various modes of capsid-mediated DNA mobilisation (GT, LT, GTA) can interact with other HGT mechanisms (transposons, integrons, ICE, plasmids) as parts of an integrated and highly active network of within- and between-species genetic exchange.

## Limitations of the study

The approach we present relies on ligation-based long-read sequencing of native DNA molecules and requires large amounts of high-quality, high-purity, high-molecular-weight DNA. The method of DNA extraction and library preparation is challenging and requires considerable amounts of freshly collected starting faecal material, limiting its application in large-scale population studies. Additionally, these factors limit the depth of long-read sequencing, resulting in some coverage patterns and size distributions being inconclusive and potentially missing certain packaging events entirely. Additional bias may be introduced by the use of CsCl density gradient[67]. Moreover, the approach we present here relies entirely on manual curation and interpretation of the results, further restricting its applications and potentially introducing bias.

Additionally, extracellular vesicles (EVs) can potentially be co-purified with capsids. This may create some difficulties in distinguishing between EV and GTA, as GTAs exclusively and non-selectively package bacterial DNA, resulting in read coverage patterns similar to those of EV DNA. While EVs are an emerging field in gut microbiome studies[68], considerable research has been conducted into bacterial EVs in marine ecosystems. It has been suggested that in marine viromes, EVs, rather than GTAs, represent the primary source of packaged non-viral DNA[30]. A broader DNA size distribution is typical for EVs[69] making it possible to distinguish between GTA and EV DNA using size analysis. However, for example, *Prochlorococcus* and *Streptomyces* EVs show the presence of discrete DNA 'bands' of a specific size[70,71].

## Methods

### Human faecal samples

Samples were obtained from three volunteers with no active gastrointestinal pathology: subject 925 (female, 48 years old); subject 928 (female, 35 years old); subject 942 (male, 39 years old). Written consents were given according to study protocol APC055, approved by the Cork Research Ethics Committee (CREC). Samples were collected in participants' homes, transported to the laboratory, and processed within 2–3 h from voiding.

### Preparation of VLPs from human faecal samples

10 grams from each of the faecal samples were subdivided into four aliquots of 2.5 g each. 20 mL of SM buffer was added to each aliquot,

and homogenisation was achieved by vortexing for 5 min. Each tube was topped up with SM buffer to a final volume of 45 ml and vortexed for an additional minute. The tubes were then centrifuged at $11,000 \times g$ for 20 min at 4 °C. The supernatant was decanted and centrifuged a second time as above. The supernatant from the four aliquots was pooled and passed through a series of syringe-mounted PES Filtropur filters; once through $0.8\,\mu m$ pore membranes and twice through $0.45\,\mu m$ pore membranes. The filtered supernatant was decanted into 10 mL polycarbonate ultracentrifuge bottles (Thermo Scientific) and centrifuged at $125,000 \times g$ at 4 °C for 2 h using a F65L-6 × 13.5 rotor in a Sorvall WX+ ultracentrifuge (Thermo Scientific). The supernatant was discarded, and the pellets from each tube were resuspended in a total of 5 ml of SM buffer, followed by additional filtration through a $0.45\,\mu m$ pore syringe filter. Five millilitres of the supernatant were then layered over a CsCl step gradient (4 mL of 3 M and 4 mL of 5 M CsCl solution in SM buffer) in 13.5 mL Quick-Seal Ultracentrifugation Tubes (Beckman Coulter). Tubes were sealed and centrifuged in an F65L-6×13.5 rotor at $105,000 \times g$ at 4 °C for 2.5 h. The VLP material produced multiple bands with two major bands visible in all three faecal samples, which were collected using a hypodermal needle and desalted by three rounds of concentration-dilution with SM buffer (50 mM Tris-HCl pH7.5, 100 mM NaCl, 8.5 mM MgSO4) on a 100 kDa MWCO Vivaspin (Sigma-Aldrich) centrifugal filter pre-equilibrated with SM buffer, each round at $4000 \times g$ at room temperature, followed by concentrating to a final volume of ~400 μl.

### Preparation of VLPs from model phage-host pairs

*Bacillus subtilis* GTA-like element PBSX particles were obtained as described in Kleiner et al. [12]. The culture was incubated at 37 °C until the optical density at 600 nm (OD600) reached 0.5. Mitomycin was then added (final concentration of 0.5 μg/ml) to induce prophage excision, and the culture was incubated for an additional 10 min before being washed and resuspended in fresh culture medium. The pellet was thus washed twice, and the culture was incubated for a further 4 h. The culture was then centrifuged, and the supernatant recovered and filtered through $0.45\,\mu m$ PES syringe filters (Filtropur).

Phages P22 and P1, as described in Kleiner et al. [12], were propagated overnight in 0.4% LB agar overlay on hosts *Salmonella enterica* LT2 and *Escherichia coli* MG1655, respectively. Top agar was collected into 3 mL of SM buffer per plate, vortexed, and mixed on a rotary mixer for 30 min. The mixture was then centrifuged at 4 °C at $3000 \times g$. The supernatant was then filtered through a $0.45\,\mu m$ syringe filter.

Phage crAss001 was propagated overnight in FAB (Neogen) 0.4% agar overlay on host *Bacteroides intestinalis* APC919/174. Filtrate was obtained from the top agar as described above.

A temperate phage from *E. faecalis* was obtained as described in Kleiner et al. [12]. *Enterococcus faecalis* VPE14089 was inoculated at 1% from an overnight culture, grown to an optical density of 0.5 at 600 nm (OD600), and induced using ciprofloxacin at a final concentration of 2 μg/mL. The culture was then grown for an additional 4 h. The culture was then centrifuged at $5000 \times g$, and the supernatant was filtered through a $0.45\,\mu m$ syringe filter.

GTA particles from *Faecalibacterium prausnitzii* were prepared as follows: a 72-h culture of *F. prausnitzii* ATCC 27766, grown in YCFA-GCSM medium, was centrifuged at $5000 \times g$ at 4 °C for 30 min. The pellet was discarded, and the supernatant was centrifuged for another 30 min. The resulting supernatant was filtered through a $0.45\,\mu m$ syringe filter. The filtrate was precipitated overnight at 4 °C in 10% w/v PEG 8000 and 0.5 M NaCl, and then pelleted at 4 °C for 20 min at $5000 \times g$. The resulting aqueous phase was further purified and concentrated to 5 mL by running twice the volume of SM buffer through a 15 mL 100 kDa Amicon column (centrifugation at $3000 \times g$ at 18 °C).

### Purifications of VLPs

Filtrates of all cultures were precipitated overnight at 4 °C in 10% w/v PEG 8000 and 0.5 M NaCl, and then pelleted at 4 °C for 20 min at

$5000 \times g$. The pellet was resuspended in 5 mL of SM buffer, and an equal volume of chloroform was added. The mixture was then vortexed and centrifuged at $3000 \times g$ for 10 min at 18 °C. The aqueous phase was collected, and the chloroform extraction was repeated a second time. VLP purification by CsCl gradient centrifugation was performed as described above. A single visible band was collected for DNA extraction. Electrophoresis of *F. prausnitzii* ATCC27766 DNA was carried out using the Genomic ScreenTape assay on an Agilent 4200 TapeStation.

## High molecular weight VLP DNA extraction

40 µL of 10X DNAse buffer (100 mM MgCl2, 500 mM CaCl2), 9 µl (2 U/µl) TURBO™-DNase, and 2 µl (1 mg/mL) RNase A (Invitrogen) were added to each 400 µl sample to remove non-encapsidated DNA and RNA. After incubation at 37 °C for 1 h, endonucleases were inactivated at 70 °C for 10 min. Two microlitres of proteinase K (20 mg/ml) and 20 µl of 10% SDS were added. The sample was incubated for an additional 20 min at 56 °C, followed by the addition of 100 µL phage lysis buffer (4.5 M guanidinium isothiocyanate, 44 mM sodium citrate, 0.88% sarkosyl, and 0.72% 2-mercaptoethanol, pH 7.0). The sample was then incubated for another 10 min at 56 °C. An equal volume of phenol:chloroform:isoamyl alcohol (25:24:1) was added and mixed gently by inverting for 2 min. Following centrifugation at $8000 \times g$ at room temperature, the aqueous phase was transferred into a sterile tube using wide-bore pipette tips. The aqueous phase was gently mixed with an equal volume of chloroform and then centrifuged at $8000 \times g$ for 10 min at room temperature. Sodium acetate (to a final concentration of 0.3 M) and 1 µL of glycogen (20 mg/mL) were added to the collected aqueous phase. An equal volume of pre-chilled (−20 °C) isopropanol was added, and after gentle mixing, the sample was incubated at −20 °C for 30 min. Following centrifugation at $17,000 \times g$ at -9 °C for 30 min, the supernatant was discarded, and the pellet was briefly washed with 1 ml of 80% ethanol. The ethanol was removed, and the pellet was left to dry before being resuspended in 50 µL of nuclease-free water overnight at 4 °C. DNA concentration was measured with the Qubit dsDNA Broad Range kit (Invitrogen).

## Total faecal DNA extraction

Total DNA was extracted from the faecal samples as described in Shkoporov et al.[72]. Briefly, 5 g of faecal sample were resuspended in 25 mL of InhibitEx Buffer (Qiagen) and aliquoted into 2 mL tubes containing a mixture of homogenisation beads (Thistle Scientific/Biospec Products): 200 µL 0.1 mm zirconium beads, 200 µL 1 mm zirconium beads, one 3.5 mm glass bead per tube. Aliquots were homogenised in the FastPrep-24 Classic bead beater (MP Biomedicals) for 30 s, chilled on ice for 1 min, followed by another round of homogenisation. The samples were incubated for 5 min at 95 °C, followed by DNA extraction using the QIAamp Fast DNA Stool Mini kit according to the protocol. DNA was quantified using the Qubit dsDNA Broad Range kit.

Selection of large (>3 kbp) DNA fragments was achieved by extraction from the excised agarose gel bands following electrophoresis. The DNA was extracted by running electrophoresis, with the excised band enclosed in a 14 kDa dialysis tubing. The DNA suspended in TAE buffer was precipitated with isopropanol as described above.

## DNA library preparation and sequencing

DNA quality was assessed using a Nanodrop 1000 spectrophotometer. DNA was considered of acceptable quality if the 230/280 nm and 260/280 nm absorbance ratios were -1.8 and 2–2.2, respectively. Sequencing libraries were prepared using the Nanopore Ligation Sequencing Kit (SQK-LSK109). Sequencing was performed on R 9.1.4 MinION flowcell (FLO-MIN110).

For short-read sequencing, 10–100 ng of DNA in 50 µL of deionised water was sheared using a Covaris M220 focused ultrasonicator

with the following parameters: time, 35 s; peak power, 50.0; duty factor, 20.0; and cycles/burst, 200. Fragmented DNA was concentrated to 15 µL using the Genomic DNA Clean and Concentrator Kit (Zymo Research). Sequencing libraries were prepared using Accel-NGS 1S Plus DNA Library Kit (Swift Biosciences). Sequencing was performed using a $2 \times 150$ bp paired-end configuration on the Illumina NovaSeq 6000. The total number of reads after quality filtration was 81–101 million per VLP DNA sample and 8–71 million per total faecal DNA sample.

## Nanopore reads from model phage-host pairs

Nanopore reads were basecalled using ont-guppy v3.4.4. FASTQ files were converted into FASTA using *seqret* in EMBOSS v6.6.0.0, and BLASTn (v.2.10.0+) was used to search against the reference genomes of bacteriophages and their bacterial hosts, with an *e*-value cut-off of 1e-20. NCBI RefSeq accession numbers are as follows: *Bacteroides intestinalis* APC919/174 substr. 8 W, NZ_CP064940.1; *Bacteroides* phage crAss001, NC_049977.1; *Escherichia coli* str. K-12 substr. MG1655, NC_000913.3; Enterobacteria phage P1, NC_005856.1; *Salmonella enterica* subsp. *enterica* serovar Typhimurium str. LT2, NC_003197.2; *Salmonella* phage P22, NC_002371.2; *Enterococcus faecalis* strain VE14089, NZ_CP039296.1; *Bacillus subtilis* subsp. *subtilis* ATCC 6051, NZ_CP034484.1. Only BLASTn alignments of 1 kb or longer were retained. In cases of ambiguous alignment of Nanopore reads to more than one genomic scaffold, the scaffold producing the greatest sum of alignment lengths with a given read was considered the correct template. Read length was determined using infoseq in EMBOSS v6.6.0.0.

## Assembly of faecal VLP DNA shotgun reads

Nanopore reads were basecalled using ont-guppy v3.4.4. No further filtration or trimming was applied. Illumina reads (Accel-NGS 1S libraries) were trimmed with cutadapt v2.8 and further trimmed and filtered using TrimmomaticPE v0.39 to remove low quality bases and reads that ended up being too short (*SLIDINGWINDOW:4:20 MINLEN:60 HEADCROP:10*). Hybrid assemblies of filtered Illumina reads and raw Nanopore reads were carried out separately for each of the samples (top and bottom VLP fractions, three donors, $n = 6$) using SPAdes v3.13.1 in *--meta* mode.

## Assembly of total faecal DNA shotgun reads

Nanopore reads were generated and basecalled using ont-guppy v6.4.6 (in MinKNOW GUI v22.12.7) using default settings for the flow cell and chemistry used. Illumina reads from Accel-NGS 1S libraries were filtered and trimmed using fastp v0.20.0 with the following parameters: *--length_required 60 --detect_adapter_for_pe --trim_tail1 2 --trim_tail2 2 --trim_front1 8 --trim_front2 13*. To remove human host contamination, reads were subjected to a Kraken2 (v2.0.8-beta) against a custom in-house database consisting of human, mouse, pig and rhesus macaque genomes (Hsap_Mmul_Sscr_Mmus_custom_db). Unclassified reads were retained for assembly. Illumina reads originating from Nextera XT libraries were filtered and trimmed using TrimmomaticPE (*SLIDINGWINDOW:4:20 MINLEN:60 HEADCROP:10*) followed by a Kraken2 search against Hsap_Mmul_Sscr_Mmus_custom_db. Hybrid assemblies of filtered Illumina reads (Accel-NGS 1S library + NexteraXT library) and raw Nanopore reads were carried out separately for each of the three faecal donors using SPAdes v3.13.1.

## Scaffold binning and selection of microbial MAGs

Scaffolds assembled from both the total community DNA and the VLP DNA (two fractions) were pooled together by faecal donor. Scaffolds were filtered to remove sequences less than 1 kb long and made non-redundant after an all-vs-all BLASTn (v.2.10.0+; *-evalue 1e-20*) by removing duplicates (shorter scaffolds having 90% of their length represented within longer scaffolds at 99% sequence identity). This procedure yielded 85,015, 42,751, and 45,265 non-redundant scaffolds (microbial and viral) for donors 925, 928 and 942. The combined

lengths of the metagenomic assemblies were 404, 254, and 158 Mb, respectively. Total community metagenomic Illumina reads (Accel-NGS 1S libraries) were mapped back to these assemblies to determine individual scaffold coverage using Bowtie2 (v2.3.5.1; *--end-to-end* mode). This was followed by converting the SAM-format alignment files into BAM files, sorting, and indexing the alignments with Samtools v1.10. Scaffold coverage data was summarised using the programme 'jgi_summarize_bam_contig_depths' from the MetaBAT2 package. Several suites of tools (MetaBAT2 dev build 2023-03-17, MaxBin2 v2.27; CONCOCT v1.1.0) were applied to bin scaffolds into candidate MAGs, with the DAS Tool (v1.1.6) being used to integrate binning results and create a non-redundant set of bins (MAGs) for each of the three faecal donors. MAGs were quality-checked using CheckM v1.2.2 and taxonomically classified with GTDB-Tk v2.3.2 (workflow *classify_wf*) using the GTDB database release 214. MAGs showing completeness of ≤50 and contamination of ≥5 were discarded, resulting in 70, 33 and 15 high-quality MAGs from donors 925, 928 and 942, respectively. Additionally, taxonomic annotation was performed on all non-redundant genomic scaffolds, regardless of the binning result, using the MMseqs2 *easy-taxonomy* workflow and the GTDB database (v214.1), which was indexed for MMseqs2 search. Functional annotation of all non-redundant scaffolds was performed using DRAM (v1.4.6, database version as of 2023-03-23)

## Selection of viral genomic scaffolds

Viral genomes/genome fragments were identified using two passes of VirSorter2 v2.2.4 on scaffolds assembled from top and bottom VLP fractions. Following the first pass (*--keep-original-seq --min-length 3000*), candidate viral scaffolds' completeness and contamination (provirus state) were assessed using CheckV v1.0.1 in the end-to-end mode. Provirus and virus sequences identified using CheckV were combined and subjected to a second pass of VirSorter2 with the following parameters: *--seqname-suffix-off --viral-gene-enrich-off --provirus-off --prep-for-dramv --min-length 3000*. The following additional filtering criteria were applied to VirSorter2-selected scaffolds to qualify as viruses: a viral gene count greater than 0, absence of host genes, a maximum score greater than 0.95, and the presence of two viral hallmark genes. Additionally, geNomad v1.6.1 (end-to-end mode) was used to identify viruses and plasmids. The combination of two tools allowed for the identification of 846, 724, and 314 complete and partial viral genomic contigs in donors 925, 928, and 942, respectively. Taxonomic classification of viruses was based on geNomad taxonomic assignments (broad level) and ICTV species-level taxonomy where possible, obtained by BLASTn (*-evalue 1e-20*) search against ICTV exemplar genomes listed in the ICTV VMR resource as of 2022-10-13. Hits were considered the same species if 85% of the scaffold of interest (combined length of BLASTn HSPs) was contained within a reference genome at a 95% sequence identity level. Additionally, virus scaffolds were searched against total community scaffolds to identify cases where virus scaffolds (one or multiple fragments, e.g., prophages) were contained within larger total community scaffolds (BLASTn, with a threshold of 99% identity and 90% of the total length aligned).

## Mapping Nanopore reads to non-redundant scaffolds

Nanopore reads of VLP DNA (top and bottom VLP fractions separately) were mapped to non-redundant scaffolds, separately, in each faecal donor, using Minimap2 v2.17-r941 and *-ax map-ont* preset. SAM alignments were converted into BAM files, sorted and indexed using Samtools v1.10. The *bamtobed* command from bedtools v2.27.1 was used to create BED files with Nanopore read alignment coordinates.

## Identification of putative GTA genes within F. prausnitzii 925_bin.142 and their homologues

The 925_bin.142 *F. prausnitzii* MAG sequence was submitted to the geNomad[73] server (https://portal.nersc.gov/genomad/), Phaster server

(https://phaster.ca/)[74,75], and Virsorter2[76] with a minimum length of 1500. Automatically identified phage regions were manually inspected for the presence of putative GTA gene features. A large terminase (CDS16 in the 52.9 kb genomic scaffold "NODE_813_length_52931_cov_10.520009") was identified by all three annotation programmes, and an upstream putative small terminase (CDS15) that was missed by the programmes was identified by manual inspection. CDS15 was submitted to Collabfold v1.5.5.5 structural prediction, with no template and three recycles. Eight additional phage-like genes were highlighted in the same 10.5 kb region (GTA I operon) by the annotation software. These included a portal, capsid, and some tail components, also identified in the MS data. As expected, no DNA replication or phage metabolism genes were identified. To identify possible homologues of these genes in related genome, a total of 16 complete and partial reference genomes of genera *Faecalibacterium*, *Gemmiger* and *Subdoligranulum* were included: ten *F. prausnitzii*, APC942/30-2 CP026548, APC918/95b CP030777, APC942/8-14-2 PRKZ0100, APC942/18-1 PRLA0100, APC942/32-1 PRLB0100, APC923/51-1 PRLD0100, APC923/61-1 PRLE0100, APC924/119 PRLF0100, ATCC 27768 PXUP0100, ATCC 27766 PXUQ0100; *F. duncaniae* A2-165 CP022479; *F. hattorii* APC922/41-1 PRLC0100; *G. formicilis* ATCC 27749 FUYF0100; *G. gallinarum* DSM 109015 JADCKC01; *Subdoligranulum variabile* DSM 15176 CP102293; and *Subdoligranulum sp*. APC924/74 PSQF0100. Translations of all coding sequences, together with translations of all coding sequences in non-redundant scaffolds from the three donors annotated by DRAM, were pooled and subjected to clustering using MMseqs2 (*cluster* command, default settings).

## *Growth curve of* Faecalibacterium

Growth measurements of the actively growing culture were taken at an optical density of 620 nm over 96 h, with readings taken at 1-h intervals and 5-s shaking performed before each measurement. The growth of the bacterium was carried out in optimal growth conditions. Readings were performed in technical triplicate.

## EM and proteomics

For negative-stain electron microscopy, 4 μL of the purified VLP sample was applied to glow-discharged Formvar/carbon 200-mesh copper grids (Agar Scientific) and stained with 2% uranyl acetate. After complete drying, the grids were imaged using JEOL 2010 TEM operated at 200 kV at York JEOL Nanocentre.

For MS analysis, 100 μL of the purified material was taken and processed through the S-TRAP protein digest procedure. Ten percent of the eluted peptides were then reconstituted in 0.1% TFA before loading onto the timsTOF HT mass spectrometer using the EvoSep One system. Analysis was performed using a data-dependent acquisition mode with elution from an 8 cm performance column, employing the 100 SPD method.

The acquired mass spectra were searched against the predicted sequences of proteins encoded by the *F. prausnitzii* ATCC 27766 draft genome (NCBI accession PXUQ0100), supplemented with viral protein sequences downloaded from the UniProt protein database. A list of common contaminants was added to this database. Searches were carried out using MSFragger version 4.1 running under Fragpipe version 22. The results were then filtered to achieve a target false discovery rate of 1% and a minimum of 2 unique peptides required for protein identification (Supplementary Data File 2).

Proteins identified by MS analysis but labelled as 'hypothetical proteins' were submitted to Collabfold v1.5.5 structure prediction, and Foldseek (GitHub release 8-ef4e960) 'easy-search' against the PDB database[77]. Top hits for each hypothetical protein were collated and used to identify other putative functions associated with phage or GTA homologues within loci other than those identified by phage annotation software. Foldseek confidence scores with an E-value cutoff of 0.1 were considered for evaluation.

## Data aggregation and visualisation

Tabular outputs from the above steps were imported into the R environment v4.4.0 and processed using a custom script (Supplementary Dataset at https://doi.org/10.6084/m9.figshare.26310658). All scaffold properties were summarised in a single table. Taxonomic assignments of prokaryotic scaffolds included in MAGs were based on GTDB-Tk, and on MMseqs2 GTDB search for unbinned scaffolds. Taxonomic assignments of viral scaffolds were derived from an ICTV exemplar genome BLASTn search, where possible, and from geNomad for the remaining scaffolds. Based aggregated data, all metagenomic scaffolds were divided into six broad categories: (1) Viruses (VirSorter2 suffix "||full" AND CheckV *Provirus* ≠ "Yes"); (2) MAGs: bacteria (scaffold part of a MAG bin); (3) MAGs: bacteria with prophages (scaffold part of a MAG bin, contains one or multiple sequence fragments of viral origin); (4) Bacteria (unbinned); (5) Bacteria with prophages (unbinned, contains one or multiple sequence fragments of viral origin); (6) Archaea (binned or unbinned).

Relative abundance of each scaffold was calculated as the number of Illumina reads aligned to that scaffold divided by the total number of quality-filtered reads for a given sample. Scaffolds enriched in the VLP fractions (Fig. S2) were identified in the following way. Firstly, a ratio of relative abundance of a given scaffold between a VLP fraction (either top or bottom) and the total community DNA in a given donor was calculated. These ratios were shown to follow a normal distribution. Secondly, VLP-overrepresented scaffolds were identified as having VLP/total community abundance ratios (as $Z$-scores) that were more than two standard deviations above the mean.

To visualise the size distribution and taxonomic origin of individual Nanopore reads, the following filtration steps were applied. Only reads with length ≥3 kb were considered. Only Minimap2 alignments of 1 kb or greater were retained. In case of ambiguous alignment of Nanopore reads to more than one genomic scaffold, a scaffold producing the greatest sum of alignment lengths with a given read was considered as the correct template. For plotting purposes, reads aligning to rare biological entities (with ≤50 reads per MAG; ≤50 reads per unbinned bacterial genomic scaffold; ≤200 reads per viral genomic scaffold) were removed.

Package gggenes v0.5.1 was used to visualise DRAM annotated genomic scaffolds, overlaid with Illumina read coverage depth (calculated by Samtools *mpileup* command, smoothed by applying a moving average), and coordinates of Nanopore reads alignments (Minimap2). The remaining plots were created using ggplot2 v3.5.1 and ggpubr v0.6.0.

## Statistics and reproducibility

Due to the nature of this study, no statistical method was employed to predetermine sample size, no data were excluded from the analyses, and the experiments were not randomised. The investigators were not blinded to allocation during experiments and outcome assessment.

Differential abundance of bacterial and viral families, defined as the total number of short reads aligned to all scaffolds within each family, was assessed across total community DNA, the top VLP fraction, and the bottom VLP fraction for all three donors using two statistical frameworks: DESeq2 and ALDEx2. The DESeq2 pipeline[78] involved estimating size factors using geometric means, fitting negative binomial generalised linear models (GLMs), and extracting differential abundance statistics for each pairwise contrast (total vs. top VLP, total vs. bottom VLP, bottom VLP vs. top VLP) using the Wald test. Taxa were considered significant at a Benjamini−Hochberg adjusted $p$-value (FDR) threshold of <0.05. In parallel, ALDEx2[79] was used to assess differential abundance based on centred log-ratio (CLR) transformed counts. ALDEx2 was run with 128 Monte Carlo instances drawn from the Dirichlet distribution, and statistical testing included both Kruskal-Wallis and GLM-based comparisons. Features were considered significant if their adjusted $p$-values (FDR) were <0.05.

To model the association between long-read sequence length and taxon abundance within families, while allowing for smooth, non-linear trends, long-read data were grouped at the family level and stratified by read length using log10-transformed intervals (bin widths of 0.1 on the log10 scale). Each read was assigned to a corresponding length bin, and read counts were aggregated per taxon, per sample, and per bin. The resulting count matrix was converted into a DGEList object for analysis with edgeR[80], and normalisation factors were calculated. A quasi-likelihood negative binomial GLM was then fit using natural spline-transformed read length (degrees of freedom = 3) as a continuous covariate, enabling flexible modelling of non-linear abundance trends across read length bins.

## Reporting summary

Further information on research design is available in the Nature Portfolio Reporting Summary linked to this article.

## Data availability

Raw sequencing data and MAGs generated in this study have been deposited in the NCBI databases under BioProject accession PRJNA1135972. Supplementary dataset, which includes MAG and genomic scaffold sequences, read mapping data, and a complete set of raw figures describing read length distributions and genomic scaffold mapping patterns of long Nanopore reads, is available at FigShare under https://doi.org/10.6084/m9.figshare.26310658.

## Code availability

A complete set of R scripts and input data needed to recreate all tables and figures in the manuscript, and a complete set of raw figures, is available at FigShare under https://doi.org/10.6084/m9.figshare.26310658.

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

## Acknowledgements

The authors thank Maria Chechik (University of York) for performing VLP imaging using TEM and the York-JEOL Nanocentre at the University of York for the instrument access and technical support; and York Centre of Excellence in Mass Spectrometry for conducting mass spectrometry analysis (supported by a major capital investment through Science City York and Yorkshire Forward with funds from the Northern Way Initiative and EPSRC [EP/K039660/1; EP/M028127/1]). The authors thank Prof. Breck A. Duerkop (University of Colorado) for providing some of the bacterial and phage strains used in this study. This research was funded by the European Research Council (ERC), under the European Union's Horizon 2020 research and innovation programme (grant agreement No. 101001684—PHAGENET). Andrey Shkoporov and Pavol Bardy were funded by Wellcome Trust Research Career Development Fellowship [220646/Z/20/Z] and Sir Henry Wellcome Postdoctoral Fellowship [224067/Z/21/Z], respectively. A BBSRC responsive mode grant funds Jason Wilson [BB/V016288/1]. Paul Fogg if funded jointly by a Wellcome Trust and Royal Society Sir Henry Dale Fellowship [109363/Z/15/A] and a BBSRC responsive mode grant [BB/V016288/1]. This research was funded in whole, or in part, by the Wellcome Trust [220646/Z/20/Z, 224067/Z/21/Z and 109363/Z/15/A]. For the purpose of open access, the authors have applied a CC BY public copyright licence to any Author Accepted Manuscript version arising from this submission.

## Author contributions

Conceptualisation, T.B. and A.N.S.; Methodology, T.B. and A.N.S.; Investigation, T.B., J.S.W. P.B., M.S., Conor H., E.V.K., C.B., M.H.; Writing—original draft, T.B. and A.N.S.; Writing—review and editing, T.B., C.B., J.S.W. P.B., B.G., P.C.M.F., C.H. and A.N.S.; Funding acquisition, A.N.S.; Resources, B.G. and A.N.S.; Supervision, P.C.M.F., C.H. and A.N.S.

## Competing interests

The authors declare no competing interests.
