## [Transparent Peer Review file · Nature Communications]

Large scale capsid-mediated mobilisation of bacterial genomic DNA in the gut microbiome

Corresponding Author: Professor Andrey Shkoporov

Version 0:

Reviewer comments:

Reviewer #1

(Remarks to the Author)

I'd like to thank the authors for their detailed responses to my queries and the additional work they have completed. I recognise the difficulty in working in pure culture with some of these isolates, but otherwise think they have done a good job addressing all my remaining concerns.

Reviewer #3

(Remarks to the Author)

The authors have adequately addressed the issues I raised. I appreciate their efforts to further clarify the limitations of their approach and the introduction of the DESeq2 and ALDEx2 analyses.

Reviewer #4

(Remarks to the Author)

I have been asked to comment on the revisions performed to address Reviewer #2's concerns, I was not an original reviewer and my feedback only concerns this review. That being said, I have read the entire manuscript and the comments of reviewers 1 and 3 for perspective.

I commend the authors for their thorough revisions. In my opinion, the authors have done a great job of addressing reviewer 2's concerns. This has been done both through revisions in text and addition of statistical analyses. I am not sure that the addition of these two statistical approaches necessarily helps due to the small initial sample size but I will not take that away from the science. I have no more comments or suggestions.

- Karthik Anantharaman
